

# Anti-holomorphic modes in vortex lattices

**Brook J. Hocking and Thomas Machon***

H.H. Wills Physics Laboratory, University of Bristol, Tyndall Avenue,
Bristol BS8 1TL, UK

* t.machon@bristol.ac.uk

## Abstract

A continuum theory of linearized Helmholtz-Kirchoff point vortex dynamics about a steadily rotating lattice state is developed by two separate methods: firstly by a direct procedure, secondly by taking the long-wavelength limit of Tkachenko's exact solution for a triangular vortex lattice. Solutions to the continuum theory are found, described by arbitrary anti-holomorphic functions, and give power-law localized edge modes. Numerical results for finite lattices show excellent agreement to the theory.



# 1 Introduction

Helmholtz-Kirchoff dynamics of point vortices [1] in a two-dimensional ideal fluid serve as a model for vortex dynamics in superfluid helium [2], Bose-Einstein condensates [3] and magnetized non-neutral plasmas [4,5]. They are related to turbulence [6] and are also of independent mathematical interest [7,8]. In an unbounded domain, there can be no finite stationary collection of vortices, instead vortex crystals [9] are found, patterns of vortices undergoing steady rigid-body motion. For large numbers of vortices, stable vortex crystals assume the form of distorted triangular lattices [10–12], and in the infinite limit a perfect triangular lattice is an exact, stable solution, as shown by Tkachenko [13,14].

Tkachenko studied the behaviour of $\ell^2$ perturbations of the infinite vortex lattice, amenable to Fourier analysis, with the result that any such perturbation may be decomposed into Tkachenko waves [14–18], plane waves where the vortices trace out elliptic paths, suggested to be related to oscillations in pulsars [19–21]. Here, the linearized dynamics of a large, but finite, lattice in an unbounded domain is studied. We show that in such a system a set of additional modes appears, described by arbitrary anti-holomorphic functions, and corresponding to power-law confined edge modes. In the continuum limit for very large (but still finite) lattices, arbitrary anti-holomorphic functions form a degenerate space of modes.

Liouville's theorem ensures that the anti-holomorphic modes do not exist in an infinite system, the magnitude of any non-constant anti-holomorphic mode is unbounded, so cannot arise as a lattice perturbation. However, their existence is revealed in the bulk spectrum, a kind of bulk-boundary correspondence. Special points in the Brillouin zone where the Hamiltonian operator $H(k)$ is defective (non-diagonalizable) have been related to edge modes [22–24]. In the vortex lattice there is a defective singularity at $k = 0$. The limit

$$\lim_{|k| \to 0} H(k), \tag{1}$$

is not unique, it yields a defective Hamiltonian, the form of which depends on the direction of approach to 0. This singularity can be attributed to the long-range nature of the vortex interactions, and the local behaviour around $k = 0$ leads to the anti-holomorphic modes.

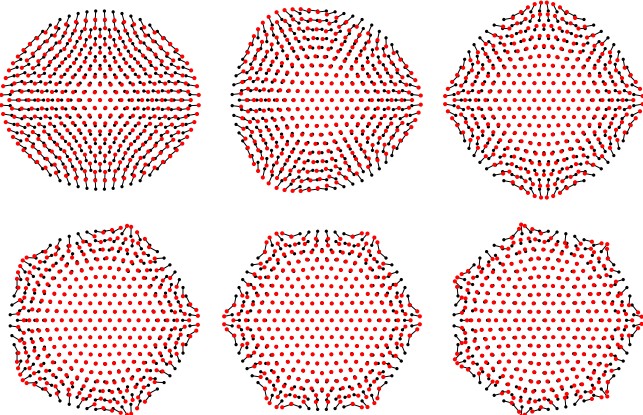

Figure 1: Anti-holomorphic modes $\bar{z}^n$, $1 \leq n \leq 6$, found numerically for a steadily rotating vortex crystal, $N = 331$. Equilibrium vortex positions given by black points, displacement under the anti-holomorphic mode given by red points.

We study these modes by developing a continuum theory around a uniform state of vorticity in a region $D$ via two separate methods, firstly by a direct approach and secondly by coarse-

graining Tkachenko's exact solution [13, 14]. The result is the equation

$$i\partial_t \overline{\Psi}(z) = -\Omega\overline{\Psi} + \frac{\Omega}{\pi}\int_D \frac{\Psi(w)}{(z-w)^2}dA_w - \frac{\Gamma}{4\pi}\frac{\partial^2}{\partial z^2}\Psi, \tag{2}$$

where $\Psi(z)$ is a complex field corresponding to the infinitesimal vortex displacement with conjugate $\overline{\Psi}$. $\Omega$ is the angular velocity of the lattice and $\Gamma$ is the circulation of the individual vortices. The first two terms correspond to a standard linearization of the ideal fluid equations, the final term accounts for the effect of the finite-vortex size, and is related (but not identical to) to the anamalous term in the hydrodynamical theory of Wiegmann and Abanov [25], as we discuss below. Our theory is linearization of the point-vortex dynamics, rather than a hydrodynamical theory [25–28].

We solve (2) under the assumption that $\psi$ describes circular oscillations of the vortices with frequency $\Omega$, yielding solutions which are approximately anti-holomorphic functions. In the remainder of the paper we investigate these modes numerically, and note that our numerical results are consistent with earlier work by Campbell [29, 30]. We show that in finite lattices of $N$ vortices, one can reconstruct anti-holomorphic modes $\psi \sim \overline{z}^n$ for $n \lesssim \sqrt{N/3}$ as linear superpositions of normal modes whose frequency is close to $\Omega$ (see Fig. 5 for an example), with the reconstruction error displaying an interesting threshold behavior (see Fig. 4). As $N \to \infty$, one recovers all anti-holomorphic modes, the error in the finite lattices can be ascribed to microscopic details leading to a splitting of the degenerate space of anti-holomorphic modes.

The anti-holomorphic modes appear as power-law confined edge waves. $\psi(z)$ anti-holomorphic corresponds to a volume-preserving vector field, and so in the continuum level preserves regions of constant vorticity. As such, the anti-holomorphic modes can be considered vortex lattice analogues of linear edge waves on the Rankine vortex, as studied by Lamb [31] (see also [32]), with the same dispersion relation. Other authors have studied edge modes, Campbell developing a continuum theory with Krasnov [17, 33, 34], with related work by Cazalilla on surface modes of vortex matter [35].

In the non-linear regime edge modes have been found to be described by the Benjamin-Davis-Ono (BDO) equation [36]. We give a preliminary study of the non-linear behaviour of the edge modes, and so not see any strong evidence of scattering between different modes, suggesting the BDO equation applies in a different regime. We also do not see the filamentation observed in the continuum system [37], whether this effect can be explicitly derived in a fully non-linear theory remains an interesting topic for future study. We note that there has been related work on statistical edge modes [38], exploiting the anomalous statistical mechanics of vortices [39]. The modes we describe are reminiscent of conformal crystals, numerically observed in vortex lattices in superconductors [40]. In compressible systems the dynamics changes [41–44], and it would be interesting to see whether the anti-holomorphic modes survive. Additionally, it would be interesting to account for effects of three-dimensionality in lattices of vortex lines [45, 46], as well as to investigate whether the antiholomorphic modes are related to the near-cyclotron-frequency waves observed in gyroscopic metamaterials [47].

## 2 Continuum theory of lattice modes

We begin with the motion of a finite collection of vortices in the plane. Let $p_\alpha = x_\alpha + i y_\alpha \in \mathbb{C}$ be the position of vortex $\alpha$, its motion is then given by

$$\dot{\overline{p}}_\alpha = \frac{1}{2\pi i}\sum_{\beta \neq \alpha}\frac{\Gamma_\beta}{p_\alpha - p_\beta}, \tag{3}$$

where $\Gamma_\beta$ is the strength of vortex $\beta$ and $\overline{p}$ is the complex conjugate of $p$. This can be written as a Hamiltonian system

$$\Gamma_\alpha \frac{dx_\alpha}{dt} = \frac{\partial H}{\partial y_\alpha}, \qquad \Gamma_\alpha \frac{dy_\alpha}{dt} = -\frac{\partial H}{\partial x_\alpha}, \tag{4}$$

with Hamiltonian

$$H = -\frac{1}{4\pi} \sum_{\alpha,\beta,\alpha\neq\beta} \Gamma_\alpha \Gamma_\beta \log|p_\alpha - p_\beta|. \tag{5}$$

(In the sum each pair $(\alpha,\beta)$ is counted twice, hence $1/4\pi$, not $1/2\pi$.) Anticipating a steadily rotating vortex crystal with angular velocity $\Omega$, we write

$$p_\alpha(t) = e^{i\Omega t} z_\alpha(t). \tag{6}$$

This gives the equations of motion in the rotating frame

$$\dot{\overline{z}}_\alpha(t) = i\Omega \overline{z}_\alpha(t) + \frac{1}{2\pi i} \sum_{\beta\neq\alpha} \frac{\Gamma_\beta}{z_\alpha - z_\beta}. \tag{7}$$

This is once again Hamiltonian, with

$$H_\Omega = \frac{\Omega}{2} \sum_\alpha \Gamma_\alpha |z_\alpha|^2 - \frac{1}{4\pi} \sum_{\alpha,\beta,\alpha\neq\beta} \Gamma_\alpha \Gamma_\beta \log|z_\alpha - z_\beta|. \tag{8}$$

A local minimum of $H_\Omega$ describes a stable vortex crystal steadily rotating about the origin with angular velocity $\Omega$. We suppose $\{z_\alpha^0\}$ is such a local minimum, then expanding $z_\alpha = z_\alpha^0 + \psi_\alpha$ to linear order in $\{\psi_\alpha\}$ yields

$$\dot{\overline{\psi}}_\alpha(t) = i\Omega \overline{\psi}_\alpha(t) - \frac{1}{2\pi i} \sum_{\beta\neq\alpha} \frac{\Gamma_\beta}{(z_\alpha^0 - z_\beta^0)^2}(\psi_\alpha - \psi_\beta). \tag{9}$$

The equations for coarse-grained vortex dynamics are now derived. Two derivations are given. The first is a heuristic physical argument. The second is a more concrete derivation based on an explicit coarse-graining of Tkachenko's exact solution for the infinite lattice. Henceforth assume all vortices have identical vorticity $\Gamma_\alpha = 2\pi$, rescaling the vorticity is equivalent to changing the timescale.

## 2.1 Derivation 1: Direct coarse-graining

We define the field

$$\chi(z,t) = \sum_\alpha \delta(z - z_\alpha^0)\psi_\alpha, \tag{10}$$

where $\delta(z - z_\alpha^0)$ is a Dirac-delta function at $z_\alpha^0$. The dynamics of $\chi$ are given by

$$\dot{\overline{\chi}}(z,t) = i\Omega \overline{\chi}(z,t) + i \sum_{\alpha,\beta,\alpha\neq\beta} \frac{\delta(z - z_\alpha^0)(\psi_\alpha - \psi_\beta)}{(z_\alpha^0 - z_\beta^0)^2}. \tag{11}$$

We regularize with some small $\sigma > 0$ with $|\sigma|$ much smaller than the lattice spacing, with the idea that we take the limit $\sigma \to 0$ and write

$$\dot{\overline{\chi}}(z,t) \approx i\Omega \overline{\chi}(z,t) + i \sum_{\alpha,\beta} \frac{\delta(z - z_\alpha^0)(\psi_\alpha - \psi_\beta)}{\sigma + (z_\alpha^0 - z_\beta^0)^2}. \tag{12}$$

Defining $\Delta(z) = \sum_\alpha \delta(z - z_0^\alpha)$, we can then find

$$\dot{\overline{\chi}}(z,t) \approx i\Omega\overline{\chi}(z,t) + i\int_{\mathbb{C}} \frac{\chi(z,t)\Delta(w) - \Delta(z)\chi(w,t)}{\sigma + (z-w)^2}dA_w, \tag{13}$$

where $z$ in the above equation is, for now, evaluated only at the points $z_\alpha^0$. We now consider the coarse-grained field

$$\Psi(z,t) = \frac{1}{A}\int_{D_z} \chi(w,t)dA, \tag{14}$$

where $D_z$ is a disk of radius $R$, area $A$, centered at $z$. $R$ is an appropriate lengthscale, to be determined. We then find the equations of motion

$$\dot{\Psi}(z,t) = -i\Omega\Psi(z,t) \tag{15}$$
$$-i\frac{1}{A}\int_{D_z}\left(\int_{\mathbb{C}} \frac{\overline{\chi}(w)\Delta(v) - \Delta(w)\overline{\chi}(v)}{\sigma + (\overline{w} - \overline{v})^2}dA_v\right)dA_w.$$

By the definition of $\chi$, there exists a non-zero $\delta$ on the lengthscale of the lattice spacing such that the numerator $\overline{\chi}(w)\Delta(v) - \Delta(w)\overline{\chi}(v) = 0$ for $|(w - z)| < \delta$. We can then safely take the limit $\sigma \to 0$. We also separate the inner integral into two pieces,

$$\dot{\Psi}(z,t) = -i\Omega\Psi(z,t) \tag{16}$$
$$-i\frac{1}{A}\int_{D_z}\left(\int_{D_w} \frac{\overline{\chi}(w)\Delta(v) - \Delta(w)\overline{\chi}(v)}{(\overline{w} - \overline{v})^2}dA_v\right)dA_w$$
$$-i\frac{1}{A}\int_{D_z}\left(\int_{\mathbb{C}\backslash D_w} \frac{\overline{\chi}(w)\Delta(v) - \Delta(w)\overline{\chi}(v)}{(\overline{w} - \overline{v})^2}dA_v\right)dA_w.$$

We now coarse-grain, assuming that $\chi(z)$ is a smooth function of $z$ varying slowly on lengthscales of order $R$. Given the lattice cell area $A_{\text{cell}}$, define the density $\rho = 1/A_{\text{cell}}$. In the coarse-grained regime we then have

$$\dot{\Psi}(z,t) = -i\Omega\Psi(z,t) \tag{17}$$
$$-i\frac{\rho}{A}\int_{D_z}\left(\int_{D_w} \frac{\overline{\chi}(w) - \overline{\chi}(v)}{(\overline{w} - \overline{v})^2}dA_v\right)dA_w$$
$$-i\frac{\rho}{A}\int_{D_z}\left(\int_{\mathbb{C}\backslash D_w} \frac{\overline{\chi}(w) - \overline{\chi}(v)}{(\overline{w} - \overline{v})^2}dA_v\right)dA_w.$$

Consider the second term. We can Taylor expand $\chi$ as

$$\overline{\chi}(v) = \overline{\chi}(w) + \sum_{a,b\geq 0} c_{ab}(v - w)^a(\overline{v} - \overline{w})^b. \tag{18}$$

We can then perform the inner integral in the second term, to leading order only the $c_{02}$ term contributes and we find

$$\dot{\Psi}(z,t) = -i\Omega\Psi(z,t) - i\frac{A\rho}{2}\partial_{\overline{z}}^2\overline{\Psi}(z,t) \tag{19}$$
$$-i\frac{\rho}{A}\int_{D_z}\left(\int_{\mathbb{C}\backslash D_w} \frac{\overline{\chi}(w) - \overline{\chi}(v)}{(\overline{w} - \overline{v})^2}dA_v\right)dA_w.$$

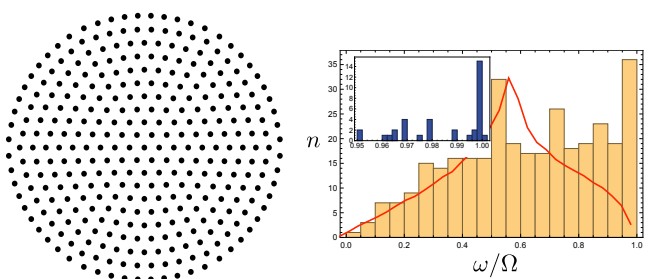

Figure 2: Left: equilibrium configuration for $s = 10$, $N = 331$ vortices. For $\omega = 1$, radius is equal to 17.44. The vortices form a distorted hexagonal lattice, note the outer circle is rotated by half a lattice consistent with Campbell and Ziff [10, 11]. Right: numerically calculated density of states for $N = 331$ (total of 331 vortices). Red line shows the continuum limit, note the absence of a frequencies near $\omega/\Omega = 1$ in this case. Inset: cluster of frequencies close to $\omega/\Omega = 1$, corresponding to anti-holomorphic modes.

We now deal with the final term. Since $\chi$ is approximately constant on lengthscales of $R$, we can approximate finding

$$\partial_t \Psi = -i\Omega\Psi - i\frac{\rho A}{2}\partial_{\bar{z}}^2\overline{\Psi} - i\rho\int_{\mathbb{C}}\frac{\overline{\Psi}(z) - \overline{\Psi}(w)}{(\bar{z} - \bar{w})^2}dA_w\,. \tag{20}$$

By choosing $A = 1/\rho$ and noting that $\rho = \Omega/\pi$ we find

$$i\partial_t\overline{\Psi} = -\Omega\overline{\Psi} - \frac{\Omega}{\pi}\int_{\mathbb{C}}\frac{\Psi(z) - \Psi(w)}{(z - w)^2}dA_w - \frac{1}{2}\partial_z^2\Psi\,. \tag{21}$$

This gives the coarse-grained dynamics.

## 2.2  Derivation 2: Small $k$ expansion of Tkachenko's solution

In this section we follow Tkachenko's work [13, 14] and consider an infinite lattice, $\Lambda$, of $2\pi$ strength vortices ($\Gamma_{mn} = 2\pi$) at sites $z_{mn}^0 = 2me_1 + 2ne_2$, with the half-periods $e_1$, $e_2 \in \mathbb{C}$, and $m$, $n \in \mathbb{Z}$. Extending the finite vortex expression to the infinite lattice leads to divergent sums for the velocity field. However, the Weierstrass $\zeta$ function has the requisite poles, so the velocity field associated to the lattice must be given by

$$\overline{v(z)} = \frac{1}{i}\left(\zeta(z; \Lambda) + f(z)\right)\,, \tag{22}$$

where $f(z)$ is an entire function. As shown by Tkachenko, if $\Lambda$ is a triangular lattice, then setting $f(z) = 0$ and using properties of the Weierstrass $\zeta$ function gives a velocity field describing a steady rigid rotation of the lattice, with angular frequency $\Omega = \pi/4\,\text{Im}(\bar{e}_1 e_2)$. Near $z = 0$ the Weierstrass zeta function satisfies

$$\zeta(z) = \frac{1}{z} - \sum_{k=1}^{\infty}\mathcal{G}_{2k+2}(\Lambda)z^{2k+1}\,, \tag{23}$$

where $\mathcal{G}_{2k+2}(\Lambda)$ is the Eisenstein series of weight $2k + 2$, in terms of modular invariants $\mathcal{G}_4 = g_2/60$ and $\mathcal{G}_6 = g_3/140$. For the triangular lattice $g_2 = 0$. (23) implies that the $(0,0)$

vortex is stationary if $f(0) = 0$. The $\zeta$ function satisfies

$$\zeta(z + z_{mn}) = \zeta(z) + 2m\zeta(e_1) + 2n\zeta(e_2), \tag{24}$$

$$e_2\zeta(e_1) - e_1\zeta(e_2) = i\frac{\pi}{2}. \tag{25}$$

and we find the velocity at each vortex is given by

$$v_{mn} = i\Omega(2me_1 + 2ne_2), \tag{26}$$

describing a rigid rotation with angular velocity $\Omega$. Concretely we set $e_1 \in \mathbb{R}$, $e_2 = e^{i2\pi/6}e_1$. Suppose now we perturb the lattice, writing $z_{mn} = z_{mn}^0 + \epsilon_{mn}$, we suppose the perturbations are in $\ell^2(\mathbb{Z}^2)$, so that

$$\sum_\Lambda |\epsilon_{mn}|^2 < \infty. \tag{27}$$

Then the velocity field can be written as

$$\overline{v}(z) = \frac{1}{i}\left(\zeta(z) + \sum_{m,n}\frac{1}{z - z_{mn}^0 - \epsilon_{mn}} - \frac{1}{z - z_{mn}^0}\right), \tag{28}$$

To first order in $\{\epsilon_{mn}\}$, the velocity of vortex $(m, n)$ may then be written as

$$\overline{v(z_{mn})} = -i\Omega(2m\overline{e}_1 + 2n\overline{e}_2) - i\sum_{(m',n')\neq(m,n)}\frac{\epsilon_{m'n'}}{(z_{m'n'}^0 - z_{mn}^0)^2} + O(\epsilon^2). $$

This yields the equation of motion (at $t = 0$)

$$\frac{d}{dt}\epsilon_{mn} = i\sum_{(m',n')\neq(m,n)}\frac{\overline{\epsilon}_{m'n'}}{(\overline{z}_{m'n'}^0 - \overline{z}_{mn}^0)^2}. \tag{29}$$

To obtain the full equations of motion we pass to the rotating frame $\epsilon_{mn} = e^{i\Omega t}\psi_{mn}$, finding

$$\frac{d}{dt}\psi_{mn} = -i\Omega\psi_{mn} + i\sum_{(m',n')\neq(m,n)}\frac{\overline{\psi}_{m'n'}}{(\overline{z}_{m'n'}^0 - \overline{z}_{mn}^0)^2}. \tag{30}$$

Now let $b_1$, $b_2 \in \mathbb{C}$ be reciprocal lattice vectors satisfying $\text{Re}(2e_i\overline{b}_j) = 2\pi\delta_{ij}$ and $BZ \subset \mathbb{C}$ the Brillouin zone. The area of the Brillouin zone is $(2\pi)^2\Omega/\pi$. Then, writing $k = k_1 b_1 + k_2 b_2 \in BZ$, we may expand $\psi_{mn}$ in plane waves as

$$\psi(k) = \sum_{m,n} e^{-i2\pi(k_1 m + k_2 n)}\psi_{mn}. \tag{31}$$

$\psi(k)$ is then found to satisfy the equation

$$\dot{\overline{\psi}}(k) = i\Omega\overline{\psi}(k) \tag{32}$$

$$-i\sum_{(m,n)}\sum_{(m',n')\neq(m,n)}\frac{\psi_{m'n'}e^{-i2\pi(k_1 m + k_2 n)}}{4((m'-m)e_1 + (n'-n)e_2)^2}. \tag{33}$$

We can rewrite this as

$$\dot{\overline{\psi}}(k) = i\Omega\overline{\psi}(k) - i\sum_{(m',n')}\beta(k)\psi_{m'n'}e^{-i2\pi(k_1 m' + k_2 n')}, \tag{34}$$

where $\beta$ is given as

$$\beta = \sum_{(m,n)\neq(m',n')} \frac{e^{-i2\pi(k_1(m-m')+k_2(n-n'))}}{4((m'-m)e_1+(n'-n)e_2)^2} \tag{35}$$

$$= \sum_{(m,n)\neq(0,0)} \frac{e^{2\pi i(k_1 m+k_2 n)}}{4(me_1+ne_2)^2} \,. \tag{36}$$

This then yields the evolution equation for $\tilde{\psi}(k)$

$$\frac{d}{dt}\psi(k) = -i\Omega\psi(k) + i\overline{\beta}\overline{\psi}(k) = H\psi(k). \tag{37}$$

$\overline{\beta}$ is a function of $k = k_1 b_1 + k_2 b_2$, where $b_1 = -4\Omega i e_2$, $b_2 = 4\Omega i e_1 \in \mathbb{C}$ are reciprocal lattice vectors (recall the lattice vectors are $2e_1$ and $2e_2$). In fact, as shown by Tkachenko [14], $\overline{\beta}$ is more readily written in terms of $\kappa = 2(k_1\overline{e}_2 - k_2\overline{e}_1) = -i\overline{k}/2\Omega$, evaluating gives

$$\overline{\beta}(\kappa) = \frac{1}{2}\left(\wp(\kappa) - \zeta(\kappa)^2\right) + \Omega\overline{\kappa}\zeta(\kappa) - \frac{1}{2}\Omega^2\overline{\kappa}^2 \,, \tag{38}$$

where $\wp$ is the Weierstrass p-function. Considering $H$ as a real $2 \times 2$ matrix, let $\pm i\omega(k) = \pm i\sqrt{\Omega^2 - |\beta(k)|^2}$ denote the eigenvalues of $H$. For the triangular lattice, $\omega$ is real and non-zero except at $k = 0$, as shown by Tkachenko, reflecting the stability of the lattice. Expanding $\beta(k)$ close to zero gives

$$\overline{\beta}(k) = -\Omega\frac{k^2}{|k|^2}\left(1 - \frac{|k|^2}{8\Omega}\right) + \frac{\mathcal{G}_6}{64\Omega^4}\left(4 - \frac{|k|^2}{\Omega}\right)\overline{k}^4 + O(k^6). \tag{39}$$

Near $k = 0$ we have $|\omega| \approx \sqrt{\Omega}|k|/2$, the Tkachenko frequency. Up to order $|k|^2$ the dispersion relation is isotropic, coarse-graining at this level should therefore yield an appropriate linearized long-wavelength theory.

Now consider the long-wavelength dynamics of $\psi(k)$ for $|k|^2 \ll \Omega$,

$$i\partial_t\psi(k) = \Omega\psi(k) + \Omega\frac{k^2}{|k|^2}\left(1 - \frac{|k|^2}{8\Omega}\right)\overline{\psi}(k), \tag{40}$$

we rewrite as

$$i\partial_t\overline{\psi}(k) = -\Omega\overline{\psi}(k) - \Omega\frac{\overline{k}^2}{|k|^2}\psi(k) + \frac{\overline{k}^2}{8}\psi(k). \tag{41}$$

Returning to real space we define

$$\Psi(z) = \frac{1}{A(BZ)}\int_{\mathbb{C}}\psi(k)e^{i(k_x x+k_y y)}dA_k \,, \tag{42}$$

where $z = x + iy \in \mathbb{C}$ and $A(BZ) = 4\pi\Omega$ is the area of the Brillouin zone, which yields

$$i\partial_t\overline{\Psi}(z) = -\Omega\overline{\Psi} + \frac{\Omega}{\pi}\int\frac{\Psi(w)}{(z-w)^2}dA_w - \frac{1}{2}\frac{\partial^2}{\partial z^2}\Psi \,, \tag{43}$$

where the integral is over the entire domain. This reproduces (21) except for the factor of $\Psi(z)$ in the non-local term. This factor gives zero in a disk domain (our focus), so the two theories agree. This arises, as Tkachenko notes [14], because the Weierstrass $\zeta$ function is not simply the sum of flow fields due to each vortex in the lattice.

## 2.3 Relation to the linearisation of Wiegmann and Abanov's theory

Here we relate (43) to Wiegmann and Abanov's theory of hydrodynamics of vortex matter [25]. We first summarise their theory. The sole free field is $\omega(r)$, thought of as $\omega = \gamma\rho$, where $\rho$ is the density of vortices and $\gamma = \Gamma/2\pi$. The vortex density is transported by the velocity field $v$,

$$\partial_t\omega + \nabla\cdot(v\omega) = 0, \tag{44}$$

which is determined from the vorticity by

$$v = (\nabla\times)^{-1}\omega + \frac{\gamma}{4}\nabla^*\log|\omega|, \tag{45}$$

where $\nabla^* = (-\partial_y, \partial_x)$, and $(\nabla\times)^{-1}$ is the inverse curl operator whose image is an area-preserving vector field. It follows that $v$ is an area-preserving vector field. Here we encounter a difficulty with linearising this theory. A disk of constant vorticity is not a steady state of this system, Wiegmann and Abanov state that one should find a number of corrections due to the $\gamma/4$ term, including oscillations in $\omega$ close to the boundary. We will neglect such difficulties here, and assume that a disk of constant vorticity is a steady state, and further neglect gradients in $\omega$.

Proceeding, we first linearize the standard ideal fluid equations, in the Lagrangian perspective, and then add in the correction term in the velocity. Suppose we have a vorticity distribution $\omega(z)$. If we perturb our vortices along some small vector field $\psi$, then transport of vorticity tells us the perturbed vorticity field will be, to linear order

$$\omega \mapsto \omega - \nabla\cdot(\psi\omega) = \omega + \nabla\times(\psi^*\omega), \tag{46}$$

where $\psi^*$ is equal to $\psi$ rotated by $\pi/2$ (or $\psi^* = i\psi$ if $\psi$ is treated as a complex function). The velocity is then seen to change, to linear order, as

$$u \mapsto u + \psi^*\omega - \nabla g, \tag{47}$$

where $\nabla g$ is a function chosen to ensure the velocity is volume preserving. We then find

$$g(z) = \frac{1}{2\pi}\int_{\mathbb{C}}\nabla\cdot(\psi^*\omega)(w)\log|z-w|dA_w. \tag{48}$$

Using the fact that $\omega$ has compact support in a disk centred at the origin, we may integrate by parts and use Stokes' theorem to rewrite this as

$$g(z) = -\frac{1}{2\pi}\int_{\mathbb{C}}(\psi^*\omega)(w)\cdot\frac{w-z}{|w-z|^2}dA_w, \tag{49}$$

finally taking the gradient with respect to $z$, and treating $\psi$ as a complex field, yields

$$\nabla g = \frac{i}{2\pi}\int_{\mathbb{C}}\frac{(\overline{\psi}\omega)(w)}{(\overline{z}-\overline{w})^2}dA_w, \tag{50}$$

now assuming the vorticity is supported on a disk $D$ with value $\omega$, we get

$$u = \frac{\omega}{2}iz + i\psi\omega - \frac{i\omega}{2\pi}\int_{D}\frac{\overline{\psi}(w)}{(\overline{z}-\overline{w})^2}dA_w. \tag{51}$$

Now, to linear order, a fluid particle initially at $z$ will be at $z + \psi$ after the perturbation. This particle will flow along the velocity field, so we obtain $\dot{\psi} = u$. Subtracting the rotational flow of $i\omega(z+\psi)/2$ gives the velocity in the rotating frame

$$\dot{\psi} = i\psi\frac{\omega}{2} - \frac{i\omega}{2\pi}\int_{D}\frac{\overline{\psi}(w)}{(\overline{z}-\overline{w})^2}dA_w. \tag{52}$$



The final step is to incorporate the additional term from Wiegmann and Abanov's theory. Their theory states that the velocity $v$ of the vortex fluid is related to the velocity $u$ of the underlying fluid flow (treating both as complex functions) by

$$v = u + \frac{i\Gamma}{4\pi\omega}\partial_{\bar{z}}\omega. \tag{53}$$

Recall our original set-up has $\Gamma = 2\pi$, and that the continuum vortex has rotational velocity $\Omega = \omega/2$. We therefore obtain the linearized theory

$$i\dot{\overline{\psi}} = -\Omega\overline{\psi} + \frac{\Omega}{\pi}\int_D \frac{\psi(w)}{(z-w)^2}dA_w - \frac{1}{2}\left(\partial_{\bar{z}}\partial_z\overline{\psi} + \partial_z^2\psi\right). \tag{54}$$

This exactly recovers the previous two results, except for the addition of the term $\partial_z\partial_{\bar{z}}\overline{\psi}$. As an addition to the velocity field, this term is of the form $\nabla^2\psi^*$, and corresponds to a non-dissipative odd diffusion term [48]. While they are related, one may distinguish $\psi$ from $\Psi$ (in (2)) as the latter is the continuum approximation of the perturbations to the discrete vortices, whereas the former is a vector field along which we perturb the vortices.

To what can we ascribe the difference between the theory we derive in (43) and the linearisation of Wiegmann and Abanov's theory? Their theory requires that the velocity field by volume-preserving, and one may check that the addition of the the odd diffusion term ensures that $\dot{\psi}$ is a volume-preserving vector field. There is, however, a slight puzzle here, as the odd diffusivity term is not the result of projecting using the standard Helmholtz decomposition (it is not curl-free). As a final point, we note that this theory also admits the anti-holomorphic modes we study in the remainder of the paper.

## 3 Anti-holomorphic modes

We now suppose the vortex lattice forms a disc $D$ of radius $R$. In this case we have

$$\Psi(z)\int_D \frac{1}{(z-w)^2}dA_w = 0, \tag{55}$$

and we find the system, using either theory,

$$i\partial_t\overline{\Psi} = -\Omega\overline{\Psi} + 4\pi\Omega\int_D \frac{\Psi(w)}{(z-w)^2}dA_w - \frac{1}{2}\frac{\partial^2}{\partial z^2}\Psi. \tag{56}$$

Numerical work [29] suggests the existence of modes that counter-rotate with the lattice at frequency $\Omega$. In regular Tkachenko waves ($\ell^2$ perturbations) the eccentricity of the ellipses traced out by the particles tends to circular as the frequency of the mode approaches $\Omega$, and so we further suppose that the trajectories traced out by the perturbed vortices are circular, *i.e* $|\Psi|$ is constant in time. Such modes are therefore solutions of the equation

$$\frac{1}{\pi}\int_D \frac{\Psi(w)}{(z-w)^2}dA_w - \frac{1}{C}\frac{\partial^2}{\partial z^2}\Psi = 0. \tag{57}$$

A first integral gives

$$\frac{\partial}{\partial z}\left(\frac{1}{\pi}\int_D \frac{\Psi(w)}{w-z}dA_w - \frac{1}{C}\frac{\partial}{\partial z}\Psi\right) = 0, \tag{58}$$

where $C = 8\Omega\pi^2$. The Cauchy-Pompeiu formula yields

$$\frac{\partial}{\partial z}\left(-\eta(z)+\frac{1}{2\pi i}\int_{\partial D}\frac{\eta(w)}{w-z}dw-\frac{\partial}{\partial z}\frac{\partial}{\partial \bar{z}}\frac{\eta}{C}\right)=0\,,\tag{59}$$

where $\Psi(z)=\frac{\partial \eta}{\partial \bar{z}}$. Now we suppose an expansion

$$\eta=\bar{z}\sum_{a,b=0}^{\infty}c_{ab}z^{a}\bar{z}^{b}\,.\tag{60}$$

We then obtain

$$\sum_{a,b}\left(-ac_{ab}z^{a-1}\bar{z}^{b+1}-\frac{1}{C}a(a-1)(b+1)c_{ab}z^{a-2}\bar{z}^{b}\right.\tag{61}$$

$$\left.+R^{2(b+1)}\begin{cases}0\,,&a-b\leq 0\\(a-b-1)c_{ab}z^{a-b-2}\,,&a-b>0\end{cases}\right)=0\,.\tag{62}$$

To solve this system of equations we identify three distinct regimes. Firstly, any anti-holomorphic function $\eta(\bar{z})$ is clearly a solution. Gathering powers of $z$ and (non-zero) $\bar{z}$ yields the equation

$$c_{a+1,b+1}=-C\frac{c_{ab}}{(a+1)(b+2)}\,,\ a\neq 0\,.\tag{63}$$

To proceed we look at all terms in (60) with a constant value of $s=b-a$, with $a\neq 0$. We first consider $s\geq 0$. Gathering terms with non-zero powers of $\bar{z}$ we use (63) and find

$$c_{i,i+s}=c_{1,1+s}\frac{(-C)^{i-1}(2+s)!}{i!(1+i+s)!}\,.\tag{64}$$

The resulting series can be summed to yield

$$-c_{0,s}(2+s){}_{0}F_{1}(2+s;-C|z|^{2})=f_{s}\,,\tag{65}$$

where ${}_{0}F_{1}$ is a generalized hypergeometric function. We now consider $s<0$. Once again we use (63) and find

$$c_{i-s,i}=c_{-s,0}\frac{(-C)^{i}(-s)!}{(i+1)!(i-s)!}\,.\tag{66}$$

Gathering terms with no powers of $\bar{z}$ yields the additional condition for $s<0$

$$-(1+s)\left(\frac{s}{C}c_{-s,0}+\sum_{b=0}R^{2(b+1)}c_{-s+b,b}\right)=0\,,\tag{67}$$

which is incompatible with (66) for $s\neq -1$. We therefore find a single additional solution, summing coefficients in (66) for $s=-1$ gives

$$\frac{1-{}_{0}F_{1}(1;-C|z|^{2})}{C|z|^{2}}=f_{-1}\,.\tag{68}$$

We therefore find a general solution of the form

$$\Psi=\gamma f_{-1}z+\sum_{s=0}^{\infty}(\alpha_{s}+\beta_{s}f_{s})\bar{z}^{s}\,,\tag{69}$$

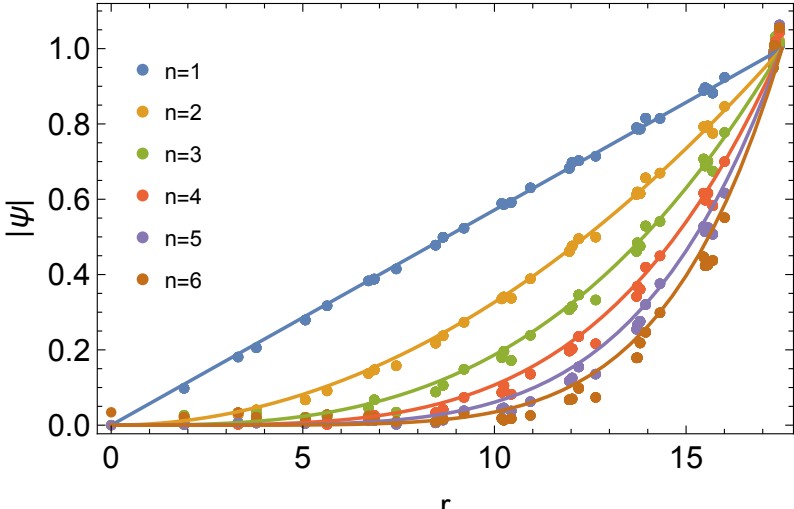

Figure 3: $|\psi|$ as a function of radial distance, $r$, for the first six anti-holomorphic modes for $N = 331 = C_{10}$ vortices. Dots are numerical data scaled so that the average value of $|\psi|$ on the outer circle of vortices ($r \approx 17$) is 1, solid lines show $r^n$, scaled to be equal to one at the outer circle. No other fitting is performed.

where $\alpha_s$, $\beta_s$, $\gamma \in \mathbb{C}$ are constants. In our coarse-grained theory we expect $|z|$ to be large compared to the lattice lengthscale, $1/\sqrt{C}$. In this limit, $f_{-1} \sim 1/(C|z|^2)$ and $f_s$, $s \geq 0$, tends to a constant (which we absorb into $\alpha_s$). We therefore find

$$\Psi \approx \sum_{s=0}^{\infty} \alpha_s \bar{z}^s, \tag{70}$$

an arbitrary anti-holomorphic function.

## 4  Numerics

For a finite vortex lattice, we define a mode of the full (non-coarse-grained) linear system with $\Psi(z) \approx \bar{z}^n$ as the $n^{\text{th}}$ anti-holomorphic mode of the vortex lattice. The $n^{\text{th}}$ mode at any given point in time gives rise to an emergent deformation structure of $n+1$ radial peaks and troughs around the edge of the system, as illustrated in Figure 1. Naively, we can expect to observe these modes only while $n \ll \Omega R^2$, where $R$ is the radius of the entire lattice, which we make precise below.

Additionally, we expect that for eigenmodes of finite lattice, the anti-holomorphic functions appear only approximately, with the degeneracy broken. In general then we expect that large but finite steadily rotating vortex lattices should have normal modes $\Psi(z)$ with the following properties (in the frame corotating with the lattice):

1. $\Psi(z)$ is approximately anti-holomorphic,

2. The normal mode counter rotates with angular frequency close to $\Omega$,

3. The paths traced out by the perturbed vortices are close to circular.

In this section we compare the prediction of the theory to numerical simulations. We consider a set of $N$ vortices in the plane all with equal circulations, $\Gamma_i = 2\pi$. We first numerically minimise the rotating-frame Hamiltonian to find an equilibrium configuration for $N$ vortices. We take $N$ to be the number $C_s$, $s \geq 0$,

$$C_s = 1 + 6\sum_{i=0}^{s} i = 1 + 3s(1+s),\tag{71}$$

which is the number of points at most $s$ steps from the origin in a triangular lattice. The vortices are initialized at the origin and the points $z_{ab} = ae^{i2\pi b/6a}$, where $a$ runs from 1 through $s$, and $b$ runs from 0 to $6a-1$. The energy (8) (with $\Omega = 1$, varying $\Omega$ simply acts as a scaling for the lattice) is then minimized. This is done using the L-BFGS algorithm in Python (scipy.optimize). An example configuration is shown for $s = 10$ in Fig. 2, the minimum is a distorted triangular lattice. We note here that Campbell and Ziff [10, 11] find the global minimum energy for configurations of $C_s$ vortices having the outer circle rotated by half a lattice constant, a result we reproduce.

Once the equilibrium configuration is found, the eigenvalues and eigenvectors of the linearized theory are computed. As expected from the theory, we find broadly two classes of modes, bulk modes with frequency $\omega < \Omega$ and a collection of modes with $\omega \approx \Omega$, the candidate anti-holomorphic modes. An example density of states is shown in Fig. 2. We note here that the density of states arising from Tkachenko's solution, defined as

$$g(\omega) = \frac{1}{A(BZ)} \int_{BZ} \delta(\sqrt{\Omega^2 - |\beta(k)|^2} - \omega) dA_k,\tag{72}$$

does not produce this excess of modes near $\Omega$, reflecting the fact that the anti-holomorphic modes do not exist in the infinite system. This collection of modes with frequency close to $\Omega$ are not a degenerate eigenspace, their frequencies are nearly, but not exactly, equal. An example is shown in Figure 5.

This cluster of modes with frequency close to $\Omega$ are our candidate modes out of which we form the 'pure' anti-holomorphic modes $\sim \bar{z}^n$. To do this we consider linear combinations of eigenmodes with frequency greater than or equal to $0.99\Omega$ and attempt to find a linear combination which best fits a deformation of the form $\psi = a\bar{z}^n$, with $a \in \mathbb{R}$. There is a second such mode, with $a$ imaginary. We do not consider this case here, but the results are similar. We now outline the fitting procedure. Let $w_k$, $k \in 1, \ldots N$ be the $N$ eigenvectors with frequency greater than $0.99\Omega$, for a given lattice of $C_s$ vortices. The eigenvectors are of length $2N$ and, for a given eigenvector $w_k$, $\overline{w}_k$ is also an eigenvector with conjugate eigenvalue, and from this conjugate pair we create a two real vectors $v_k = \text{Re}\, w_k$ and $v_{k+1} = \text{Im}\, w_k$. Then the displacement at some site $i$, located at $z_i$, will be

$$d_i = \sum_{k=1}^{N} f_k v_{ki},$$

where $f_k \in \mathbb{R}$ and $v_{ki}$ the $i$ component of $v_k$. Now let $\psi_n = a\bar{z}^n$ be the anti-holomorphic mode. Then we define the total error as

$$\epsilon_{n,s} = \sum_{i=1}^{C_s} |d_i - a\bar{z}_i^n|^2.\tag{73}$$

This is then minimised over the constants $f_k$ to obtain the fit. The error $\epsilon_{n,s}$ can then be used as a measure of the goodness of fit. In order to be comparable across different modes and

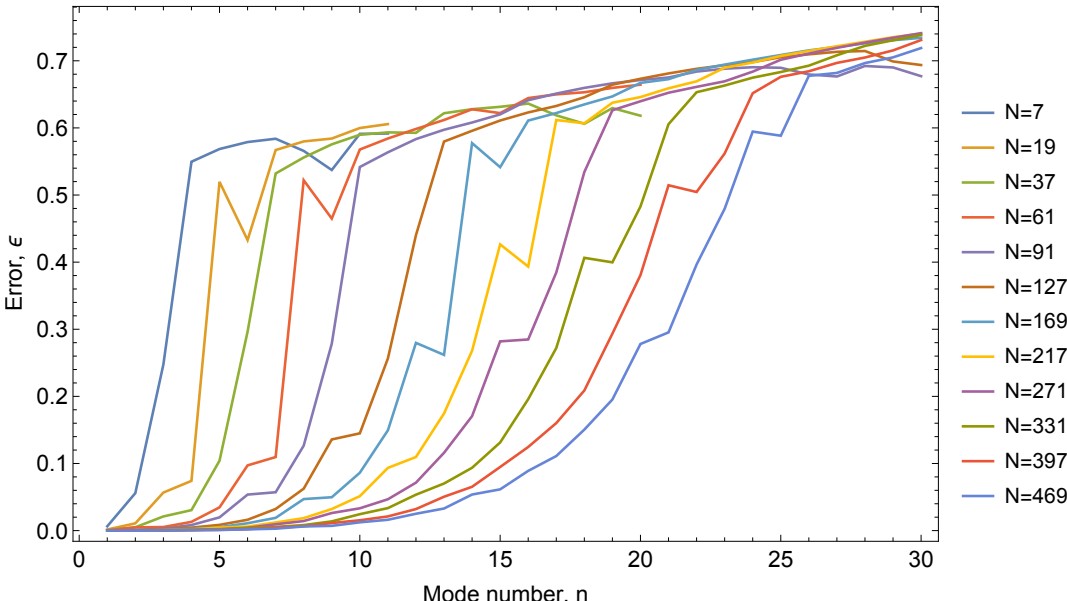

Figure 4: Error, $\epsilon$, defined in (73), for fitting the anti-holomorphic mode $\bar{z}^n$, for lattices of size $N$. For $N$ large there is a sustained region of low error, corresponding to realisable anti-holomorphic modes. The error grow with $n$ and is observed to reach a threshold value before increasing linearly. The threshold is found to grow as $N^{1/2}$.

system sizes we must choose $a$ appropriately. Initially we chose $a$ so that setting all the $f_k$ to be zero would yield an $\epsilon_{n,s}$ of 1. This choice is given by

$$a_0 = \left(\sum_{i=1}^{C_s} |\bar{z}_i|^n\right)^{-1},$$

however, it was found, empirically, that using $a = a_0 C_s^{1/3}$ produced an excellent data collapse, shown in Figure 4. We note that this is only required to compare fits across different lattice sizes, and does not affect the fits themselves.

Figure 4 shows the minimised error as a function of lattice size and mode number. We observe that the error rises quickly until a threshold at which point it rises slowly, in a roughly linear fashion. For reasonably large lattice sizes there is a substantial region of low $\epsilon$, which corresponds to anti-holomorphic modes in the lattice that are observable. If we consider a value of $\epsilon$ of approximately 0.1 as indicative of low error, then we find that the threshold grows as $\sim \sqrt{N}$, and as a rule of thumb we find, for $C_s$ vortices, that the first $s \sim \sqrt{N/3}$ modes can be accurately reproduced. For example, for $N = 61$ vortices, the first 4 anti-holomorphic modes, $n = 1, 2, 3$ and $4$ may be observed.

Fits found in this fashion for 331 vortices are shown in in Fig. 1 for $1 \leq n \leq 5$. To assess the accuracy we plot the magnitude $|\psi|$ as a function of radius $r$, which should scale as $r^n$. This is shown in Fig. 3, and we find excellent agreement for $1 \leq n \leq 6$, for higher $n$ the correspondence begins to break down.

In the coarse-grained theory, anti-holomorphic functions are a degenerate eigenspace with frequency $\Omega$. In the real system this degeneracy is broken by microscopic details, and we find a collection of modes with frequencies just below $\Omega$. This is illustrated for the $n = 1$ mode with $N = 61$ in detail in Figure 5. As linear combinations of powers of $\bar{z}$, the eigenmodes appear as power-law confined chiral edge modes of the vortex lattices.

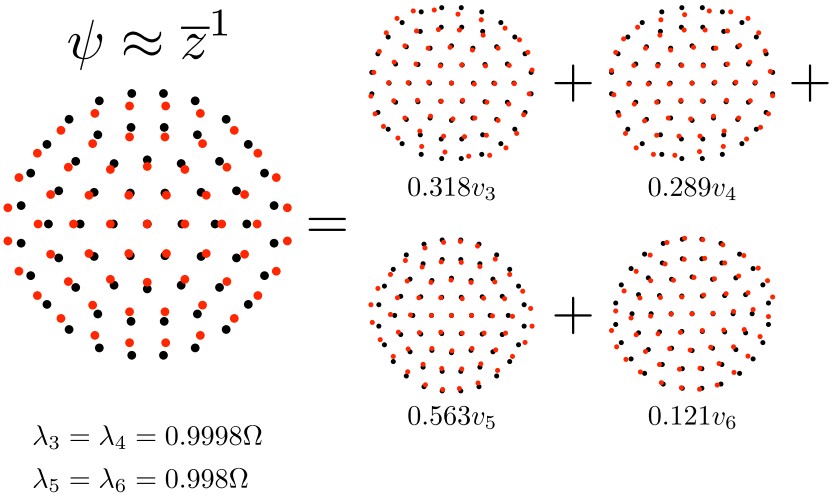

Figure 5: For $N = 61 = C_5$, the mode $\psi \approx \bar{z}^1$ is constructed as a linear combination of the third through sixth eigenvectors, whose corresponding eigenvalues are close, but not exactly equal, to $\Omega$. If we were instead to fit $a\bar{z}^1$, where $a \in \mathbb{C}$ was arbitrary, the relative magnitude of the four contributions would change, but the same four eigenvectors would produce the mode. Note that the first and second eigenvectors correspond to uniform translations ($\psi \sim \bar{z}^0$) and are always exact eigenvectors with eigenvalue $\Omega$.

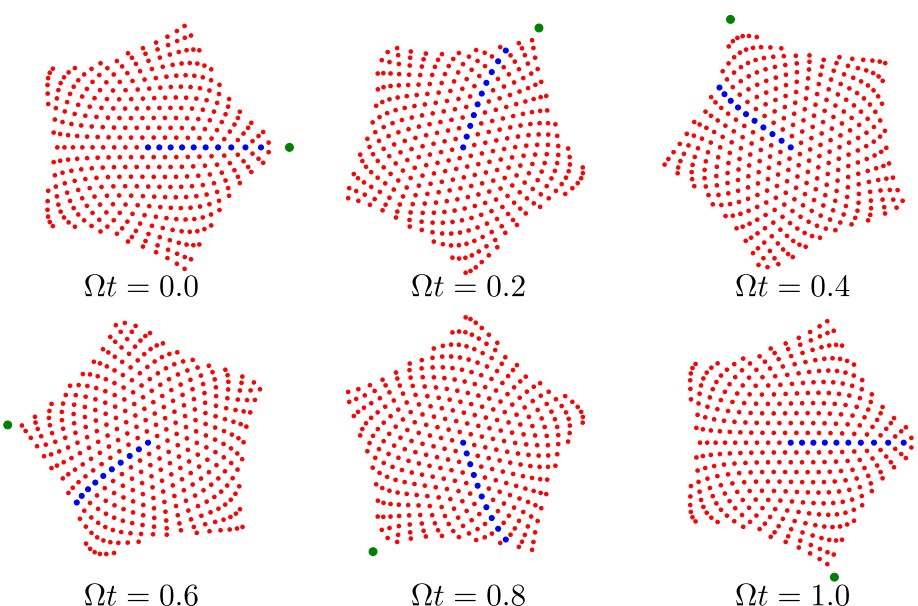

Figure 6: The $n = 4$ mode. A reference set of vorticies are highlighted in blue, and orbit the center with angular frequency $\Omega$. A reference deformation peak is indicated by the green dot, and orbits the center with angular velocity $\frac{n}{n+1}\Omega$.



How do these modes appear in the lab frame? In the rotating frame, each vortex rotates about a lattice point with angular frequency $-\Omega$. However, in the lab frame the whole system rotates with angular frequency $\Omega$, as such each vortex is displaced from the lattice by a time independent vector. In particular, from this we can find that the maxima of radius appear at $\theta$ obeying

$$\theta = \frac{n}{n+1}\Omega t + \frac{2\pi}{n+1}p\,,\tag{74}$$

where $p \in 1, 2, ..., n$. We then see the second term defines the $n + 1$ peaks, and the first term tells us that this emergent structure rotates with frequency

$$\frac{n}{n+1}\Omega\,,\tag{75}$$

as depicted in Figure 6. This difference in frequencies is analogous to the difference in frequencies associated to solar days and sidereal days as a planet orbits a star; as the lattice point orbits the center, the direction of the perturbation required to create a deformation peak is also shifted. The analogous phenomenon in the continuum case is described by Deem and Zabusky [32].

## 5 Nonlinear behaviour

How do these modes behave beyond linear order? In an ideal fluid, the nonlinear behaviour of a perturbation to a circular vortex blob leads, ultimately, to filamentation and singularity formation, as investigated by Dritschel [37]. In the point vortex system, filamentation is supressed by the microscopic lengthscale introduced by the vortex lattice [36]. To begin to understand the nature of the nonlinear behaviour of these modes we give a preliminary numerical investigation, shown in Figure 7, which shows the nonlinear behaviour of the $n = 1$ mode in a lattice of size 61. We investigated other lattice sizes and mode numbers with similar results. The numerical integration was performed using scipy solve_ivp , with the Radau implicit solver. The mode shown in Figure 7 is remarkably coherent, lasting well beyond 50 periods (this timescale is obviously reduced as the perturbation becomes larger). We can project the displacement of the nonlinearly evolved lattice back on to the eigenvector spectrum, also shown in Figure 7. Beyond 50 periods the mode begins to degrade, as shown by the image of the lattice at 100 periods.

How can we understand the resulting behaviour? One possibility is via a nonlinear wave equation governing the edge of the blob, such as the Benjamin-Davis-Ono (BDO) equation [36], where the anti-holomorphic mode of order $\bar{z}^n$ corresponds to the $n$th edge mode. In this case we would expect scattering between anti-holomorphic modes of different wavenumbers. Although our numerical study of the nonlinear behaviour is only preliminary, we do not observe any significant scattering between anti-holomorphic modes. It is likely that the regime in which the BDO equation was derived is not applicable here, indeed, one suspects that the solitons described in Ref. [36] may be observed in a finite lattice by adding an extra vortex at the edge of the lattice, and observing it rotate around the edge. Such a soliton is of fundamentally different character to the anti-holomorphic modes we describe. We note also that the middle lattice in Figure 7 illustrates a shearing of the lattice as the mode begins to degrade. This is perhaps consistent with Wiegmann and Abanov's theory [25], which suggests that the boundary of a uniform blob of vorticity will rotate at an altered rate, due to gradients in $\omega$.

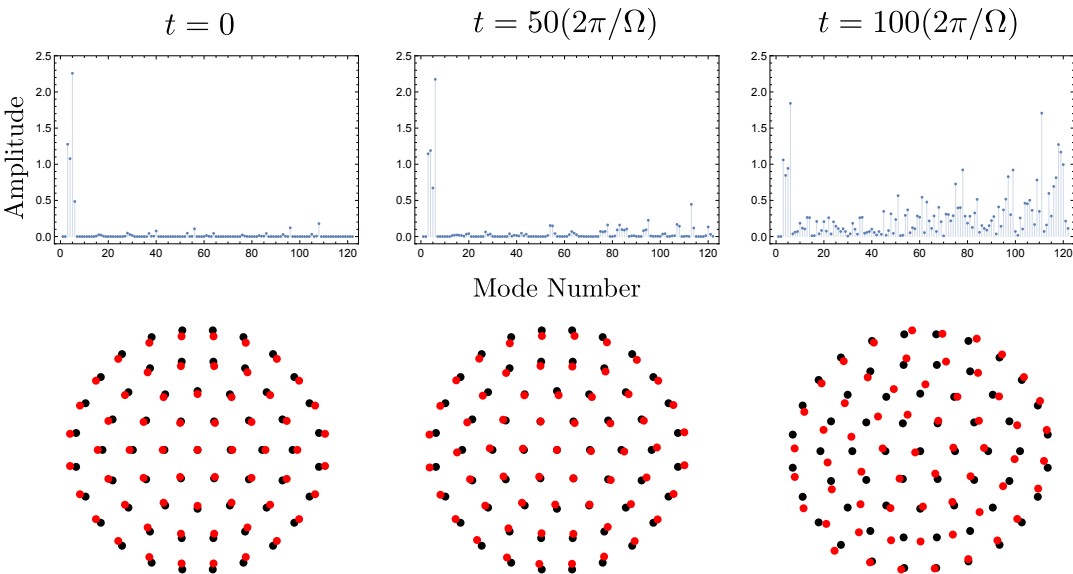

Figure 7: Nonlinear behaviour of the $n = 1$ mode in a lattice of size $N = 61 = C_4$ with $\Omega = 1$ (the radius of the lattice is of order 7). Left: an initial perturbation moves the vortices (red) from their equilibrium points (black) along an anti-holomorphic mode. In this case, the perturbation is by $0.05\bar{z}$, giving a relative perturbation of 5%. Centre and right: the perturbation evolved nonlinearly after 50 and 100 periods respectively (in the rotating frame). After 50 periods the $n = 1$ mode retains a degree of coherency, after 100 periods it is no longer coherent. Top: eigenvector spectrum for the three lattices. The four prominent peaks corresponding to $v_3$ through $v_6$ are clearly visible (see Figure 5).

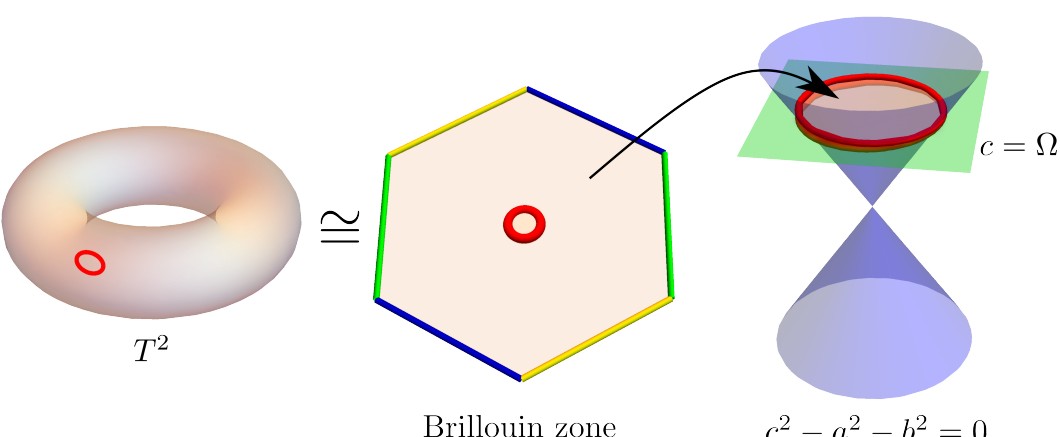

Figure 8: Left/Middle: The Brillouin zone with a puncture at $k = 0$. A loop (red) around the puncture, when shrunk to the origin, is mapped to a loop winding twice around the surface of the cone. The rest of the Brillouin zone is not defective and so does not map to the surface of the cone. Right: The set of defective hamiltonians form a cone (blue), with the frequency $\Omega$ dictating the height (green).

# 6   Bulk-boundary correspondence

The non-constant anti-holomorphic modes we describe cannot appear in an infinite system, by Liouville's theorem they are not bounded, and so cannot be realized as perturbations of a lattice in any physically meaningful sense. However, as demonstrated by the numerical results, in a large but finite system they appear, albeit approximately. If one considers the non-constant anti-holomorphic modes as (power-law confined) edge waves, this can be considered as a kind of bulk-boundary correspondence. In fact, their existence can be ascribed to the $k = 0$ singularity in Tkachenko's solution. The Hamiltonian nature of the original vortex system ensures that, as a real linear operator on $\tilde{\psi}(k)$, $H(k) \in \mathfrak{sl}(2, \mathbb{R})$, the Lie algebra of traceless $2 \times 2$ matrices. $H(k)$ has a singularity at $k = 0$, the bulk spectrum of the vortex lattice therefore defines a map from the punctured Brillouin zone into $\mathfrak{sl}(2, \mathbb{R})$, as

$$H(k) : BZ \setminus \{0\} \to \mathfrak{sl}(2, \mathbb{R}). \tag{76}$$

We may write a general element of $\mathfrak{sl}(2, \mathbb{R})$ as

$$X = \begin{pmatrix} a & b + c \\ b - c & -a \end{pmatrix}. \tag{77}$$

The set of defective Hamiltonians with this symmetry can be identified with the cone $c^2 - a^2 - b^2 = 0$, null vectors in Minkowski space $\mathbb{M}^{2,1}$. Excluding the origin in $\mathfrak{sl}(2, \mathbb{R})$ gives the set of defective Hamiltonians the homotopy type of two disjoint circles. We now consider the explicit form of the spectrum for the vortex lattice, near $k = 0$ we may write

$$
\begin{aligned}
H(k) = &-\Omega \begin{bmatrix} \sin 2\theta_k & \cos 2\theta_k - 1 \\ \cos 2\theta_k + 1 & -\sin 2\theta_k \end{bmatrix} \\
&+ \frac{|k|^2}{8} \begin{bmatrix} \sin 2\theta_k & \cos 2\theta_k \\ \cos 2\theta_k & -\sin 2\theta_k \end{bmatrix} + \mathcal{G}_3 O((|k|/\Omega)^4).
\end{aligned}
\tag{78}
$$

The overall rotation of the lattice forces $c = \Omega$ in (77). Coupled with Tkachenko's dispersion relation $|\omega| \approx \sqrt{\Omega}|k|/2$, we see that as a small loop around $k = 0$ is shrunk to the origin in the Brillouin zone, the image under the map $H(k)$ is a loop winding around the cone of defective matrices, with winding number 2. The local structure of this singularity is sufficient to reproduce the anti-holomorphic modes.

## Acknowledgements

We thank T. Gavrilchenko, J.M.F. Gunn, J.H. Hannay, Y. Hu, and R.D. Kamien for helpful discussions. This work was supported by the EPSRC through grant EP/T517872/1.

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
