# Peer review of "Anti-Holomorphic Modes in Vortex Lattices"

_SciPost Physics, doi:SciPost Phys. 14, 080 (2023)_

## Round 1 · Referee Report · Anonymous (Referee 2) · 2022-8-15

Report

I thank the authors for improving the paper.

Before I recommend to publish this paper in SciPost Physics, I have two additional comments:

1) Could the authors identify a term in their equation (2) which encodes elastic effects of the vortex lattice. I am surprised that the comparison with the WA hydrodynamic theory of a vortex fluid (that obviously has no elasticity) leads to essentially the same equation (55).

2) In a recent work [Phys. Rev. B 106, 024308 (2022)] , surface modes near cyclotron frequency were identified in two-dimensional crystals with broken time-reversal symmetry. Although in that paper vortex crystal were not considered, I would like to encourage the authors to investigate if these cyclotron surface waves are related to the anti-holomorphic surface modes discussed in the manuscript.

  • validity: -
  • significance: -
  • originality: -
  • clarity: -
  • formatting: -
  • grammar: -

Author:  Thomas Machon  on 2022-11-20  [id 3050]

(in reply to Report 1 on 2022-08-15)
Category:
answer to question

We thank the referee for their comments.

1) The elasticity of a vortex lattice is interesting, and the general elasticity of 'conformal materials' is a topic that could receive more interest. The vortex lattice has no bulk modulus and the dynamics are first order (there is no dynamical matrix formulation), so it is not clear that elasticity is the correct way to think about the system. However, any effect of the lattice itself comes from the \partial_z^2 term (note that, for example it permits arbitrary dilations).

2) This is an interesting comment, particularly the near-cyclotron nature of the modes linked. We have added a comment suggesting this as an area for future study.

---

## Round 1 · Referee Report · Anonymous (Referee 3) · 2022-9-8

Strengths

The manuscript makes an important contribution to the theory of edge waves on a chiral vortex system by presenting a theory for these edge waves in different parameter regimes and using a different approach to what has been reported in previous works in this field. The revised manuscript now clearly places the work within context and therefore advances our understanding of the different edge modes that can exist in this system.

Weaknesses

Some typographical errors that need to be addressed that I have listed in my report.

Report

The author has responded to all the key points in my initial report and I am happy that these key points have now essentially been addressed.

However, I have spotted a number of typos which should be corrected although many of these are minor. Upon correction, I would be happy to recommend the paper to be published.

Requested changes

In the revised manuscript, there are a few minor points that should be clarified/rectified:

  1. When introducing Eq. (2), define the overbar as complex conjugate at that point, and define the region of integration. Also, below Eq (2), Gamma is the circulation and not the vorticity.

  2. Below Eq. (2) it is stated that the last term of that equation corresponds to the anomalous term in [25] but then after Eq. (55) it is stated that another additional term (\partial^2_{z^z} \psi^) is the one that corresponds to odd viscosity. Are these statements consistent?

3.Correct sentence at end of page 1 `Our theory is a linearization ...'

  1. On page 4, three length scales appear, R, delta and sigma. It would be helpful to explicitly state the assumption made in terms of the relative magnitude of these length scales in relation to one another.

  2. Before Eq. (45), it seems lowercase \gamma is also circulation and not vorticity which when multiplied by vortex number density would then give the vorticity omega.

  3. In going from Eq. (49) to Eq. (50), I assume this is simply integration by parts.

  4. After Eq. (51), state ` ... omega is supported on a disk D with constant value omega ...'

  5. Sentence after Eq. (54) is incomplete.

  6. The final equation (55) is in terms of lowercase \psi whereas in previous sections and introduction uppercase \psi was used. Could the notation for the quantity \psi be made consistent across these expressions?

  7. Correct sentence after Eq. (73) ... `In order to be comparable ....'

  8. Page 8, second column, paragraph beginning with Fits found ...', reference is made to Fig. 1 but I believe this should be Fig. 3 and 1 \le n \le 6 (not 5). Also a couple of lines later there is repetition of... we find excellent agreement ...'

  9. In Fig. 7, the labels on the axes of the figures are difficult to read. I would recommend using a larger font size.

  • validity: high
  • significance: high
  • originality: high
  • clarity: good
  • formatting: excellent
  • grammar: good

Author:  Thomas Machon  on 2022-11-20  [id 3051]

(in reply to Report 2 on 2022-09-08)
Category:
answer to question

We thank the referee for their comments. We have addressed all the minor points in revision.

---

## Round 1 · Author Response

We would like to thank the referees for their careful and thorough reading of the manuscript. In revision we have made substantial additions to the manuscript, largely to the numerical aspects of the work, which are now equal in scope than the theoretical aspects. The major additions include: a comparison of our work with the theory of Wiegmann and Abanov, a more thorough examination of existence of anti-holomorphic modes in finite lattices, and a preliminary study into the nonlinear behaviour. Additionally we have made a number of smaller changes, some of which address the referees' minor concerns, and others elaborate on additional aspects of the work.

Main revision 1: Relation of our theory to other theories

We first clarify the nature of our derivation. Of the two derivations we give, the coarse-graining of Tkachenko's exact solution is the strongest. The principles behind it are well-established in condensed matter physics. For example, identical techniques (Fourier transforming, assuming $\psi(k)$ is localised near the origin in $k$-space, then replacing the Brillouin zone with all of $\mathbb{R}^2$ and inverse Fourier transforming) may be used to establish the linear elasticity of two-dimensional crystals (interacting via an arbitrary potential). In this sense, we are applying condensed-matter techniques, and perspective, to the problem, rather than ideas from fluid dynamics.

The referees asked about the nature of the $\partial_z^2$ term. This can be considered as a correction to linearised ideal fluid behaviour coming from the finite vortex size. In revision we have added a comparison to the theory of Wiegmann and Abanov (WA). Linearizing their theory in analogous fashion gives the same $\partial_z^2$ term and an additional chiral odd diffusive term. In general we would not expect the two theories to agree. Our theory is a coarse-grained linearisation, whereas the WA theory coarse-grains first, finding a hydrodynamic theory where the velocity field is required to be volume preserving. One can check that the chiral odd diffusive term ensures that $\dot \psi$ is a volume-preserving vector field. Interestingly, it does not correspond to the standard Helmholtz projection (i.e. adding a gradient), which remains something of a mystery. The regime of the WA theory is different to our work. $\psi$ may be much larger than the lattice spacing in the WA theory (see further comments in the reply to referee 1). As a curiosity, it turns out that if the additional two WA terms are halved in magnitude, then by some algebraic manipulation, one can show that the extra term does indeed correspond to the Helmholtz projection. We thoroughly checked both theories and were not able to find such a factor (though we did find a typo in Equation 22 of the WA 2013 PRL (ArXiv version)). We note that both theories permit anti-holomorphic modes.

A second aspect of this is the possibility of finding a nonlinear version of our work. There are two issues with this. Firstly, we would have to first coarse-grain, then linearise, which would require dramatically different techniques, beyond the scope of our work. Secondly, the nonlinear numerics suggest that a nonlinear wave equation is not appropriate, so it is not clear what one would be looking for.

Main revision 2: Nonlinear behaviour of the anti-holomorphic modes

In revision we have performed some nonlinear studies of the behaviour of the anti-holomorphic modes. We note that the modes are remarkably coherent, with reasonably large perturbations lasting for many tens of periods (we have not done a detailed study here, which is beyond our scope). Additionally, the decay appears to be into low-frequency bulk modes, not scattering into other anti-holomorphic modes. As such, it appears that a nonlinear wave equation for the edge is not a good approximation in this regime. This is particularly relevant to the Benjamin-Davis-Ono equation, where the dispersive term should lead to scattering. We suggest that the regime for the BDO equation is of a much larger vortex lattice, with much smaller perturbations. In fact, one can take a stable vortex lattice and add one extra vortex to the edge, which should orbit. Such a vortex is most likely corresponding to a BDO soliton.

Main revision 3: Breakdown of the continuum theory

In revision we have performed a detailed study of the breakdown of the anti-holomorphic modes as the lattice size grows. We found some interesting results. We construct an error function, measuring how well the mode can be produced in a given system. This error is shown to quickly grow with the mode number, until a plateau, where it then grows more slowly. The onset of the plateau appears to grow with the square root of the number of vortices. The results indicate that one can observe, roughly, anti-holomorphic modes up to $n \approx \sqrt{N/3}$ modes in lattices of $N$ vortices.

We now reply in detail to the points made by the referees.

Referee 1:

  1. The main coarse grain approximation is the old and well-known problem of `contour dynamics?. see e.g. D. G. Dritschel, The repeated filamentation of two-dimensional vorticity interfaces J. Fluid Mech, 194, 511 (1988); I understand that it corresponds to the first two terms in the rho of (20). It is known that contour dynamics develop filament instabilities. I believe the authors must include some discussion on this issue.

1 reply. The referee is right to point out the relevant work of Dritschel, and we have discussed this in revision (see end of introduction, nonlinear behaviour section, and discussion in the reply above). The main coarse-grained approximation is closely-related, but not identical to, contour dynamics, which is concerned with regions of constant vorticity, where dynamics can be determined purely in terms of the boundary. The field $\psi$ is a perturbation out of this regime, so contour dynamics are no longer applicable. Nevertheless, the theories are closely related. As an interesting aside, we believe the BDO equation can be obtained from Dritschel's work, truncating (6) in his paper to quadratic order in $\eta$ (one may have to be careful choosing the correct frame of reference to get the BDO equation exactly, but one notes that the linear integral term is precisely the circular Hilbert transform. Additionally (6) refers to the spherical case, which is somewhat different to the planar case, as Dritschel notes, for the planar case one must use $2\eta + \eta^2$, which makes the algebra messy). This BDO equation would {\em not} be the same as the one derived by Bogatskiy and Wiegmann.

  1. I understand that the last term in (20) is related to the lattice structure. This term would be absent if the vortices form a liquid rather than crystal. Is it correct? Is this term sensitive to Tkachenko's moduli of the lattice? For example, how does this term depend on the lattice, say, square or triangular?

2 reply. The infinite square lattice is an unstable equilibrium (shown by Tkachenko), so we would not expect to observe it in any reasonable scenario. Nevertheless, one can study the linearised theory about the square lattice. In this case, the linearised theory would be different -- in particular it would have a second-order differential operator that was not isotropic, that is the lattice could be detected on the coarse-grained level. In the triangular lattice case, the coarse-grained theory is isotropic (this is a phenomenon repeated in two-dimensional phonons). Regarding the liquid phase -- one can consider the theory of Wiegmann and Abanov, which is coarse-grained and does not assume any lattice structure. As we show, the linearisation of that theory contains a similar term to the one we derive, so the additional term is perhaps better characterised as due to the finite size of the vortices.

  1. Does this term stabilize the development of filaments? 3 reply. As our entire formulation is based around the linearisation of the dynamics, it is difficult to answer this question. One would have to formulate a nonlinear theory (a la Wiegmann and Abanov) and study that, which is beyond our scope. We note that their theory also presents difficulties, as even the steady state is not easy to obtain (and is not a uniform $\omega$). Finally, we note that the nonlinear numerics dynamics suggest that the edge modes may decay into bulk modes that could be invisible under coarse-graining, which may stabilise any edge perturbations in a continuum description.

  2. In the case of the liquid (the first two terms in the rho of 20) the linear edge modes had been extensively studied in papers by Crowdy. A discussion of a relationship with these works will be helpful;

4 reply. We must apologise, but we were unable to find such a study by Crowdy. He has many papers studying more complex situations such as~\cite{crow1,crow2}, but we were not able to find a study of the simple disk. If the referee could provide a detailed reference, we would be happy to add a discussion. We note that Lamb studied the linear behaviour of the Rankine vortex, and we have added a comment.

  1. It will be desirable to formulate the equation for the edge mode as an equation in terms of the real coordinate along the edge. And when it is done what is the relation between the last term in 20 and the dispersive term of the Benjamin-Ono equation discussed in [34].

5 reply. This is addressed in the main points above.

Referee 2:

  1. In Introduction the authors state:``In the laboratory frame these modes are zero-frequency, coupled with the overall rotation of the lattice leads to a time dependent deformation structure rotating in the same direction as the lattice, but at a slower frequency.'' I do not understand this sentence and encourage the authors to clarify it. Why the frequency is slower?

1 reply. This is explained in Fig. 6 (revised manuscript) or Fig. 4 (original manuscript), as well as in the surrounding text. We agree the original text was not well written, and on review we feel it did not add to the introduction, so have removed it.

  1. In the end of "Introduction" the authors suggest that their edge modes might be related to the Kohn mode discussed previously in Refs [17,42] in vortex crystals. As far as I understand, the Kohn theorem is valid also in infinite systems without any boundary, so why do the authors see any connection to their edge excitations?

2 reply. This comment was speculative, and for clarity we have removed it. However, for the benefit of the referee, in Ref. 42 (original manuscript) the authors find an additional set of modes that are circular (like the anti-holomorphic modes) and have a non-zero frequency at $k=0$ (like the anti-holomorphic modes, which are all at $k=0$), and one may speculate that they are related.

  1. In Fig 2, what is meant by ``(totalling 331)''?

3 reply. 331 vortices, this has been amended.

  1. After Eq. (22) what are $g_2$ and $g_3$? 4 reply. $g_2$ and $g_3$ are the modular invariants of the lattice, we have added a comment to this effect.

  2. If for the triangular lattice $\alpha = 0$, why the function $f(z)$ is at all before Eq. (26)? 5 reply. This was poor writing, we have removed $f(z)$ in the revision.

6 In the section "Anti-holomorphic modes" the authors start from the ansatz (53). How come that the expansion coefficients depend on $|z|^2$, Does not that contradict the recursion relation (56)? 6 reply. Using equation numbers from the original, (53) is the most general Taylor series form for $\eta$ (we can assume at least one power of $\overline{z}$ since we only care about $\Psi = \partial_{\overline{z}} \eta$). Then we substitute this general form into (52), which gives a new power series in $z$ and $\overline{z}$, which must vanish. The resulting equations come from setting each of the coefficients to zero, there is no issue.

  1. In Eq. (63) it appears there is a typo $s\to i$.

7 reply. Fixed.

  1. Could the authors provide some details on the linearized solution of the microscopic model discussed in ``Numerics''?

8 reply. We hope the referee is satisfied by the revised content on this topic.

  1. This is beyond the scope of the paper, but can the bulk-boundary correspondence from the end of the paper be extended to the compressible superfluids, where the Tkachenko mode is quadratic?

9 reply. An excellent suggestion, additionally one may think about effects arising from three-dimensionality (vortex lines) also beyond the scope, however we have added a comment regarding such ideas.

Referee 3: The main points of referee 3 are addressed above.

  1. Above Eq.(50), it is assumed that the perturbed vortex trajectories can be assumed to be circular. Although the assumption sounds plausible, how can this be better justified?

4 reply. In regular Tkachenko waves ($\ell^2$ perturbations) the eccentricity of the ellipses traced out by the particles tends to circular as the frequency of the mode approaches $\Omega$, which one can see by examining the Brillouin zone picture. We have added a comment to this effect.

5 It is stated that the solutions obtained in Eq. (62) correspond to neglecting the term involving the second derivative with respect to z. What is the physical effect of this term on the solutions obtained for higher anti-holomorphic modes where this term would become more important?

5 reply. We have removed this comment in revision, as it was not helpful for the manuscript. However, this limit is taking the individual vorticities to zero, recovering the continuum case. In the solutions we find, of the form (\alpha_s +\beta_s f_s) , f_s is constant at large distances, and drops to zero at the origin, which happens on a length scale determined by the lattice. While it is true that for high modes this term is more relevant, as f_s grows more slowly, the pure anti-holomorphic solution, determined solely by alpha_s, still exists.

  1. Phrasing of sentence following Eq. (63) is a little unclear. 6 reply. Reworked.

  2. It is stated that the red curve (rather than red line) shown in Fig. 2 is obtained from the continuous theory and does now contain the edge modes. I was unsure what equation was used to recover this curve so it would be useful to state this explicitly. 7 reply. This is now given explicitly. The function is calculated numerically.

---

## Round 1 · List of Changes

1. Added a comparison to the theory of Wiegmann and Abanov
  2. More thorough discussion of error in observing modes in finite lattices
  3. Added a figure showing how a given mode is constructed out of eigenvectors
  4. Added a preliminary study of the non-linear behaviour of the modes, with comparison to related studies.
  5. Minor changes and comments in line with referees' comments.
  6. Other minor changes and edits.

---

## Editorial Decision

published